# The Horizontal Rain-Cell Span and Wind Impact on Multisite Diversity Scheme in a Tropical Region during El-Niño and La-Niña

**DOI:** 10.3390/s23146424

**Published:** 2023-07-15

**Authors:** Fazdliana Samat, Mandeep Singh Jit Singh, Abdulmajeed Al-Jumaily, Mohammad Tariqul Islam

**Affiliations:** 1Pusat Sains Angkasa, Institut Perubahan Iklim, Kompleks Penyelidikan Universiti Kebangsaan Malaysia (UKM), Bandar Baru Bangi 43600, Selangor, Malaysia; 2Department of Electrical, Electronic and Systems Engineering, Faculty of Engineering and Built Environment, Universiti Kebangsaan Malaysia (UKM), Bandar Baru Bangi 43600, Selangor, Malaysia; tariqul@ukm.edu.my; 3Department of Signal Theory and Communications, Universidad Carlos III de Madrid (UC3M), 28911 Madrid, Spain; abdulmajeed@tsc.uc3m.es

**Keywords:** atmospheric attenuation, signal degradation, tropical climate

## Abstract

Site diversity is the most effective way to recover a signal lost during heavy downpours, especially in tropical regions since other mitigation techniques such as adaptive power control and code modulation may be unreliable during such. Duplicated links at diverse sites are deployed, and the least-attenuated signal of either site will be routed to the prime site for further operation. Since the deployment is costly, a diversity-gain model is used to estimate the appropriateness of selected sites. Diversity gain is known to depend on site-separation distance and elevation angle and, optionally, baseline angle and signal frequency, based on the region of research. In addition to these factors, the horizontal rain-cell span and the wind’s impact on the gain are ongoing investigations, especially in tropical regions. This article presented the rain analysis from the year 2014 to mid-July 2017 at eight sites in the Gombak and Sepang districts of Malaysia to investigate the dependency relevancies. The rain rates were then used to predict the attenuation using the ITU-R P.618-13 rain-attenuation model, and the inter- and cross-district gain characteristics were evaluated. The observation of diurnal rain during the northeast seasons yielded that the northeast wind stimulates intense rain at locations along its direction, thus, extending the horizontal rain-cell span to 15 km distant from a host. Meanwhile, sites located at 5 km distant, slightly perpendicular to the wind direction, and from 90° to 180° from due north of the host, experience less rain. The baseline angle variation establishes nonimpact to the gain and lengthening the site-separation distance presented equal chances to the shorter span towards diversity-gain increment. The research outcome is necessary to formulate a more reliable diversity-gain model to be used in the industry.

## 1. Introduction

Satellite communication has been in our lives for years on our planet Earth. The existence of globally connected communication facilitates gathering information from other countries beyond the boundaries of the terrestrial network. The main issue in satellite communication appears during rainy events when the signal is degraded by the droplets, especially at above 10 GHz frequency [1]. The severely affected area is the tropical region due to its heavy rain climate throughout the year [2]. Therefore, the research focusing on techniques to mitigate the rain impact on the signal is significant for satellite communication industries in the area. Several mitigation techniques have been proposed and implemented, such as adapting power transmission according to the rain attenuation level of the receiver; however, maintaining the high value for a long time is costly and inefficient [3]. Another technique is the modification of the signal coding and modulation to adapt to the attenuation level, but transmitting the changed code would compromise the signal quality [4]. Moreover, the satellite signal in the tropical regions could have been lost before it reaches the receiver due to severe rain impact [5]. Hence, the former techniques are perhaps no longer practical to be implemented considering those occurrences.

Diversifying the site is one of the proposed ways to help recover the satellite signal, called site diversity. The Earth station receiving the signal is multiplied and they are located at different locations from the reference site, the host. The separation distances are at least beyond the horizontal rain-cell span [6]. The signal received at the diverse site is considered as the backup to the host. As the rain-induced attenuation is getting worse, the least-attenuated signal from the diverse site would be routed to the host and used in the operation. Therefore, the backup site should be selected properly, which gives many benefits to the host [7,8].

Hodge, in his article, proposed that diversity gain be used to measure the performance of a chosen diverse site because of its practicality than the improvement factor [9]. The gain is the function of attenuations at both sites, given a certain percentage of time exceedance probability. Due to the dearth of satellite signal measurement availability, Hodge proposed the empirical model for gain prediction using datasets measured in temperate regions. The model presented four gain dependencies: the signal frequency, site-separation distance, antenna-elevation angle, and the orientation angle between the baseline and the azimuth line [10]. Panagopoulos et al. [11] presented that the Hodge model was insensitive to the increment of the site-separation distance. The author proposed that the gain increases in logarithmic correlation to the distance and is not saturated at a certain length, as presented by Hodge.

In tropical regions, Jane et al. investigated seasonal variations of site diversity implementation in Nigeria [12]. The rainfall data were obtained from the Meteorological Agency-NIMET (2010 to 2014) at four sites and deduced the rainfall rate and distribution using the Chebil and Moupfouma models, respectively. The rainfall data were categorized according to four Nigeria climate seasons, December to February, March to May, June to August, and September to November. The ITU-R model was used to estimate the rain attenuation at each site; then, the joint attenuation (the least attenuation between paired sites) was derived from the rain attenuation distribution. The author concluded that the site diversity is best implemented during the March to May season (high diversity gain compared to other seasons). Yeo et al. investigated the dependencies’ impact on the diversity gain and reported that the wind direction may affect the attenuation depth in a certain locality, thus, indirectly influencing the diversity gain [13]. Nonetheless, the analyses showed that there is only a slight difference in the statistical gain collected yearly. Further investigation was conducted, using an assumption of 20 GHz signal frequency and 50° antenna-elevation angle, in Singapore districts. Yeo concluded that the local horizontal rain-cell extent is 15 km, and the gain does not depend on frequency and baseline angle, hence, proposing a new unlike-Hodge model structure that excludes both parameters [14]. Semire et al. developed a Hodge-like model with different coefficients that suit the tropical region [15]. Despite agreeing with Yeo that the frequency and baseline angle have less impact on the gain, those parameters are included in Semire’s model [16]. Hence, there is a lack of information on the wind’s effect on diversity gain, especially in tropical regions, such as Malaysia, which receives two monsoon seasons in a year.

Sites at different locations experienced unlikely similar attenuation to the other site due to the inhomogeneity of rainfall types [17]. The spatial horizontal rain-cell span has been reaching at least 15 km, investigated in tropics by [18], and 10 km in the temperate region [19]. The site-separation distance has also been investigated using micro rain-cell site diversity, by which the predefined threshold classifying the smaller rain-cell (in meters) span was monitored [20], and multiple site diversity, which normally has a longer distance (in kilometers) between each site [21]. In the former, Shukla et al. simulated the rain attenuation at seven sites, assuming a 30 GHz operating frequency and antenna elevation of 61°. The author concluded that in the Ahmedabad of India, the horizontal rain-cells span during the intense monsoon spell exhibited a smaller area, at around 100–300 m only; hence, attenuation dissimilarity could be achieved at least beyond 500 m of distance [22], while the latter investigated multiple site diversity in Guayaquil assuming a satellite signal of 20 GHz and an elevation angle of 67.7° at four different places. The outcome revealed that the longer the site-separation distance, of at least 17 km, the attenuation gets more unlikely. These investigations showed that the longer the distance, the better. However, since Hodge presented otherwise, there is a need to investigate those that influence the diversity gain; besides the four gain dependencies (including the site-separation distance), the wind direction and the rain cell -span, specifically in the tropical region, for a more reliable gain prediction model developed in this area.

This article presents the analyses of rain events from 2014 to mid-2017 (42 months) in Sepang and Gombak Districts of West Peninsular Malaysia during the northeast seasons to observe the impact of wind during the monsoon spell and the minimum span of site-separation distance that contributes to high gain. The observation is also to validate the horizontal rain-cell span proposed by Yeo et al. and indirectly determine the influence of baseline angle on the diversity gain. This investigation is essential to observe the gain dependencies of a diversity-gain prediction model. The ITU-R 618-13 rain attenuation model was used to predict the attenuation depth of the sites. The diurnal rain event was also presented to investigate the rain-type relevancy and time-series occurrences from one site to another in each district. The list of sites’ locations, the site-separation distance of each, and the model used for gain comparisons are presented in the Materials and Method section. The Results section presents the rain rate and rain attenuation CCDF graph, and the sampled rain-event duration to support the suggestion. The Discussion part elaborates on the behaviors of gain observed when pairing the site, and the impact of the northeast wind on the gain. Lastly, a conclusion was drawn.

## 2. Materials and Methods

Rain gauges were used in the measurement of the rain-density readings at eight sites, four sites each in Gombak and Sepang districts. The sites chosen are located near the physical earth station (gateway) at Rawang (101.5546° E 3.30364 ° N) and Cyberjaya (101.6584° E 2.9356° N) town, respectively. In Sepang, Ladang Bukit Cheeding (LBC) is located at 101.34° E 2.51° N, Ladang Teluk Merbau (LTM) at 101.41° E 2.51° N, Paya Indah (PI) at 101.37° E 2.52° N and Puncak Niaga Selangor (PNS) at 101.41° E 2.54° N. While in Gombak, Country Homes (CH) is located at 101.31° E 3.19° N, Taman Bukit Rawang (TBR) at 101.35° E 3.19° N, Taman Desa Kundang (TDK) at 101.31° E 3.18° N, and Taman Garing Utama (TGU) at 101.33° E 3.20° N. Figure 1 and Figure 2 demonstrate the location of each site in the Sepang and Gombak districts, respectively. The distance between sites ranged from 1.87 km to 14.13 km, as shown in Table 1.

The rain-density sampling was in one-minute intervals, recorded by the data logger at each site. The measurement was conducted by the Department of Irrigation and Drainage (DID), Selangor, Malaysia, from 1 January 2014 to the middle of July 2017, using rain gauges. The tipping bucket is 200 cm in diameter, connected to a logger to record the rainfall in 1-min time intervals. The tipping-bucket accuracy percentage is ±2% for readings of 25 mm/h to 200 mm/h and ±3% for readings of 300 mm/h to 500 mm/h. The readings were analyzed using Microsoft Excel in *.csv* format. The rain rate of one-year readings was obtained by segregating the rain-density occurrences. Each one-minute rain value was converted to hourly form by multiplying the corresponding amount by 60; hence, the unit was in millimeters per hour (mm/h). For example, if the rain density were 0.1 mm, converting it into the rain rate would become 6 mm/h. The frequency of occurrence of each rain rate was accumulated, then divided by the total available readings and multiplied by 100 to get the percentage of time exceedances. The procedure was repeated for each year of available measurements at each site.

The rain rates at a percentage of time exceedance 0.01% ($R_{0.01})$ of each year at sites in Sepang and Gombak were extracted. Since deploying the antenna at each station incurs costly expenses, the rain attenuation at each site was predicted using the ITU-R P.618-13 rain attenuation model [23] with each site’s rain rates at $R_{0.01}$ as input to the model. The ITU-R formula was preferred because of its simplicity and is widely used by researchers in this field to model rain attenuation. Furthermore, the comparison is consistently made at the same instantaneous probability of time exceedance so that the accuracy is preserved. The ITU-R model estimates the attenuation *A* at the percentage of exceedance *p*, denoted as $A_{p}$ by using the predicted attenuation derived from $R_{0.01}$, which is denoted as $A_{0.01}$, as shown in Equation (1). The $A_{0.01}$ was derived from the specific attenuation, which consists of the rain rate at a 0.01 percentage of time exceedance and the coefficient k and α, and $L_{eff}$, which is the effective path length, as shown in equation (3). Those coefficients could be calculated according to ITU-R P.838 [24], given the signal operating frequency and antenna polarization. The elevation angle is denoted as θ. The details on the ITU-R P.618-13 formula can be found in the referred documentation.
(1)$$A_{p}={A_{0.01}\left(\frac{p}{0.01}\right)}^{-(c-\beta \left(1-p\right)\sin\theta )}$$
where in:(2)$$c=0.655+0.033\ln\left(p\right)-0.045\ln\left(A_{0.01}\right)$$
(3)$$A_{0.01}=kR_{0.01}^{\alpha }L_{eff}$$

Rain attenuation was simulated with the assumption that each site is the domestic/terminal antenna receiver with diameters less than 2.4 m. The antenna parameters were assumed to be set with resemblance to the physical gateways station nearby, which in detail can be found in [25]; signal frequency of 20.2 GHz from MEASAT-5 at an orbital location of 119.5° E with an elevation angle of 68.8°. The rain height $h_{r}$ was set to a value suggested by the recommendation ITU-R P.839, which is 4.86 km. The coefficients *k* and *α* are 0.0981 and 0.9831, respectively, assuming vertical polarization. The latitude input in the model varied according to the location of the sites.

Malaysia is a tropical climate country, located near the Equator, and receives two seasons in a year, the northeast monsoon from November to December of the current year, until March of the consequent year, and the southwest monsoon, from late May to September of the current year. The former brings heavier rainfall than the latter [26]. Normally, during the northeast season, the wettest month is in November, thus inducing the worst subsequent rain-induced attenuation in that year [27]. The signal is expected to be down for most of the time during this month because of the heavy downpour. Considering this situation, the rain rate in November of each site was extracted for each year, and inputted into the ITU-R model formula, to observe the predicted rain attenuation during the month. 

The probability of time exceedance at 0.01% is an acceptable quality of service for satellite communication defined by ITU-R. Since the 0.01% rain rate at each site during November of each year varies, some sites experienced the highest and the lowest rain rate compared to other sites. Therefore, the outcome of predicted attenuation derived from the extracted rain rate was compared. The site with the highest rain rate was set as a host, and it was paired with the rest of the sites in the district, the diverse site. The comparison was made to deduce the joint attenuation, and the diversity gain consequently. 

Figure 3 defines the terminology for determining the gain, denoted as $G_{D}$ [28]. The percentage of time exceedance is denoted as *P*. The diversity gain was obtained by taking the difference between the single-site attenuation at a certain percentage of time exceedance, $A_{s}(P)$ and the joint attenuation of the same percentage of time exceedance $A_{j}(P)$, as presented in (4). Joint attenuation is the minimum value of instantaneous attenuation between the host and diverse sites; denoted as $A_{D}\left(P\right)$. The observed is at the same percentage of time exceedance, as in (5). The diversity gain was extracted from the joint attenuation distribution and single site attenuation distribution, consistent with the procedure in [12].
(4)$$G_{D}\left(P\right)=A_{s}\left(P\right)-A_{j}\left(P\right)$$
(5)$$A_{j}\left(P\right)=\min\left(A_{s}\left(P\right),A_{D}\left(P\right)\right)$$

The cross-district rain events were also observed to investigate the effect of lengthening the site-separation distance on the diversity gain. The distance from each place in Gombak was measured to the areas in Sepang, with the details of their respective values shown in Table 2. The least-measured direct distance was 44.82 km, while the longest was 55.75 km. Due to a similar distance range, a gain comparison was made between the site with the highest rain rate throughout the measurement year with the rest of the sites. The purpose of the comparison was to observe the impact of lengthier site-separation distance on the diversity gain.

The gains extracted from the pair of the highest and lowest rain rate of a site diversity scheme in Sepang and Gombak were compared with four empiric gain models, the Hodge, Panagopoulos, Yeo, and Semire. These formulas were recreated using Microsoft Excel for Microsoft 365 (version 2306). The Hodge, Panagopoulos, and Semire models resemble each other in their structure except for the coefficients while Yeo is a bit different. In the former, the gain dependencies on site-separation distance, $G_{d}$, frequency, $G_{f}$, elevation angle, $G_{\theta }$, and baseline angle, $G_{\varphi }$ are highlighted as the factors that contribute to the gain calculation. For the sake of simplicity and to avoid much redundancy, the formula of Hodge is listed in (6) to (11) while Panagopoulos and Semire’s model structure can be referred to in their respective articles. The notation d is the distance, A is the attenuation value, f is the frequency in GHz, θ is the antenna-elevation angle in degree, and φ is the baseline angle, also in degree.
(6)$$G_{d}=a(1-e^{-bd})$$
where in:(7)$$a=0.64A-1.6(1-e^{-0.11A})$$
(8)$$b=0.585(1-e^{-0.098A})$$
(9)$$G_{f}=1.64e^{-0.025f}$$
(10)$$G_{\theta }=0.00492\theta +0.834$$
(11)$$G_{\varphi }=0.00177\varphi +0.887$$

Therefore, the final predicted diversity gain, $G_{D}$ is the multiplication of each gain.
(12)$$G_{D}=G_{d}G_{f}G_{\theta }G_{\varphi }$$

The Yeo structure of diversity-gain calculation is simpler because the frequency and baseline angle are not included as one of the factors because of the previously mentioned reason, presented in (13).
(13)$$G_{D}=(-0.78+0.88A)(1-e^{-0.18d})(1+e^{-0.14\theta })$$

The input for model was the same for frequency, *f* which was 20.2 GHz and the elevation angle, the θ was 68.8°. The distance, *d* was varied according to the site being compared and so the baseline angle. The details on the sites’ parameters can be found in the Discussion section.

## 3. Results

Three parameters have been analyzed: the rain rate, the consequent attenuation, and the diversity gain of each site. Figure 4 shows the rain rates of each site in the Gombak and Sepang districts from January 2014 to mid of July 2017 according to the percentage of time exceedance. From the figure, the rain rate at 0.01% of time exceedance at CH was 80 mm/h, TDK was 85 mm/h, TBR was 92 mm/h, and TGU was 88 mm/h during the measured duration. The rain rates at 0.01% of time exceedance were also extracted in Sepang districts, whereby site LBC was 76 mm/h, LTM was 73 mm/h, PI was 144 mm/h, and PNS was 77 mm/h. It was noticeable that the PI graph was isolated and, hence, received the highest rate than the other sites. CH, TGU, TBR, and TDK in Gombak and LBC, LTM, and PNS in Sepang were in a similar pattern of rain-rate range and graph despite their site-separation distance being over 40 km (refer to Table 2).

Rain rates at a point of location are not the only cause of signal attenuation because a series of rain along the propagation path may accumulate the impact; however, it may be a premier source. Therefore, a correlation can be imposed that the local rain rate is directly proportional to attenuation, such that the higher the local rain rate, the higher the attenuation, as defined in the ITU-R rain attenuation model formula. Therefore, with the fixed parameters mentioned in Section 2, the site’s rain rates with its latitude were taken as inputs to the rain attenuation modeling. Figure 5 shows the estimated rain attenuation at sites in Sepang and Gombak districts during the measured duration using ITU-R P.618-13 rain attenuation model. PI showed the highest attenuation value among others, which was 45.63 dB during 0.01% of time exceedance. The high rain rate of PI, which was 144 mm/h, as shown in Figure 3, for 0.01% of the time exceedance is expected to cause the high attenuation. Other sites showed a similar range of rain attenuations because of their rain-rate resemblances. Sites in Gombak, the TDK, CH, TBR, and TGU were predicted to experience attenuations in the range of 32.49 dB to 35.28 dB with rain rates ranging from 80 mm/h to 92 mm/h, at 0.01 percentage of time exceedance. The sites in Sepang, except for PI, the LTM, PNS, and LBC predicted attenuations ranging from 30.87 dB to 31.87 dB, with rain rates ranging from 73 mm/h to 77 mm/h during the same percentage of time concerned. While the attenuation at CH, TDK, TGU, and TBR in Gombak exhibited higher values than those in LBC, LTM, and PNS in Sepang, the difference was only 1.62 dB to 3.41 dB, despite their far distance (over 40 km) and differences in geographical location.

As a result of the highest rain rates in PI, it was expected that the signal received by this site was severely impacted by the rain. Therefore, considering PI as the host, it was paired with other sites to infer the diversity gain, and the value was compared with the respective site-separation distance. The calculated diversity gains from PI to the site nearby of the same district and the site outside of the district are presented in Table 3. The distance of more than 40 km is the site outside the district, while the remaining are the ones nearby. From the table, the highest gain was observed at a site separation of 7.63 km, between PI and LTM, which was 14.77 dB, while the lowest gain was observed at a 50.17 km distance between PI and TBR, which was 10.35 dB. This information revealed that the rain rate of a distant location may not differ from the host; the closer one may also not necessarily be similar. Therefore, the receiver located at PI should not necessarily be connected to a far diverse site to get the strong satellite link, hence reducing the unnecessary cost of connecting the two sites. Meanwhile, the best location to connect is at the host’s east which is the LTM, less than 10 km far (site-separation distance only 7.63 km).

Practically, the diverse site’s signal is only routed to the host during a certain time frame on a rainy day (e.g., broadcasting), especially when the host experiences heavy downpours, which causes severe attenuation and probably a signal loss. Therefore, the diurnal rain event at a specific time during the month of heaviest rain, such as November, is worth observing. As presented before, in Sepang, the site PI and LTM were receiving the highest and lowest rain rate (at 0.01 percent time exceedance), measured in two years; from 2014 to 2016. The same goes for sites in Gombak, which demonstrated that TBR and CH were receiving the highest and lowest rain rate at 0.01 percent time exceedance from 2014 to 2016. Thus, the rain rate in November of each year at the highest and lowest rain-rate sites was extracted and displayed in Figure 6. Figure 7 shows the rain rates in November for the rest of the sites, TDK and TGU of Gombak and LBC and PNS of Sepang, during the equal duration. The segregation of the graphs is to demonstrate that each site’s rain rate during the November spell may vary despite its highest and lowest rain rate during the years. For example, according to Figure 6, it was observed that PI did not receive the highest rain rate in November 2014; instead, for Sepang, Figure 7 presented that PNS received the highest rain rate at 0.01 percent of time exceedance. TBR in Gombak was not receiving the highest rain rate, but CH, exhibited in Figure 6, during the November spell in 2014. Therefore, the subsequent discussion follows.

The rain event within the district was observed to determine the rain cell’s span. In Gombak, during November 2014 in Figure 6, CH and TBR was receiving 120 mm/h and 31 mm/h rain rates at 0.01 percent time exceedance, respectively, while TDK experienced the highest rain which was around 147 mm/h, and TGU was 122 mm/h, as presented in Figure 7. Therefore, by assuming TDK (the highest rain rate) as a host, CH and TGU are at its north and northeast, separated by 1.85 km and 5.24 km in distance, respectively. While TBR was experiencing the least rain rate during November 2014 for 0.01 percent of time exceedance, the site is 7.63 km far northeast of TDK. It is at the east of CH, and southeast of TGU, separated by 7.43 km and 4.14 km, respectively.

Despite its close distance from those sites (especially TGU, compared to the Sepang sites), TBR received a far lower rain rate during the same percentage of time exceedance. Therefore, the host’s selection of the nearby diverse site with less rain rate and possibly less consequent attenuation was TBR, then the CH and TGU. Considering sites at CH were also experiencing lost signal because of the high rain rate, it also needs backup from nearby sites. Hence, in the CH viewpoint, TDK experienced more signal attenuation due to its high rain rates, so it should be left unselected; thus, the nearest location with a stronger signal was also TBR. This sample showed that the rain cells are different at a site-separation distance of at least 4.14 km to the east or southeast of the host during this time.

In Sepang, Figure 7 exhibited that during November 2014, PNS received the highest rain rate for 0.01 percent of time exceedance, amounting to 142.5 mm/h, while the lowest was at LTM (Figure 6), with 26.5 mm/h. Therefore, considering PNS as a host, the nearest site with a stronger signal was the LTM, located right in its south at 5.56 km apart. LBC is located at the west of PNS, 12.96 km away but was experiencing 87 mm/h, as well as PI, with 111 mm/h, more than that of LTM. Similarly, considering using PI as the host, the most appropriate diverse site during this period apparently is the LTM, at its east 7.63 km apart. It is because the signal propagates through the low rain rates at LTM and is likely to experience less attenuation. LTM may receive a better signal than the LBC and PNS, which are 6.68 km and 8.28 km away, respectively, from PI but with heavier rain. Thus, the condition presented by this sample suggests that the rain differed by at least 5.56 km at a site location either in the east or south direction of the host.

Hence, the November 2014 diurnal rain-event sample was analyzed in a specific time frame for each circumstance in Gombak and Sepang. Since the highest rain rate at 0.01 percent of time exceedance during this time in Gombak was TDK, its rain density in the time series was extracted. The data were taken directly from the raw source, which exhibited the timing format in 24 h with extension ‘00’ as its seconds. For example, time 3.01 p.m. was presented with ‘150100’, with ‘15’ as ‘3’ in 24-h format, ‘01’ as its minute, and ‘00’ as its seconds. From the observation of the raw data in time series, TDK received the highest rain, which amounted to 3.8 mm, on 21 November 2014. Therefore, the rain events in the time series at sites TGU, TBR, and CH were observed during the corresponding day and specific time; from 15:00 to 15:50 (50 min), as demonstrated in Figure 8.

Figure 8 presented the rain event at four sites in the Gombak district within the equivalent time frame from 15:00 p.m. to 15:50 p.m. The rain patterns of each site are similar to one another but in different time intervals. TDK started to receive rainfall starting from 15:15 p.m., with the rain density rapidly increasing until its peak at 15:38 p.m., and reducing about twelve minutes later at around 15:50 p.m. When TDK received the highest rain density of 3.8 mm (228 mm/h) at 15:38 p.m., CH received 1.4 mm, TGU, 1.3 mm, while TBR received 0.9 mm, which corresponds to 84 mm/h, 78 mm/h, and 54 mm/h, respectively. However, each site reached its peak rainfall at different times; CH at 15:34 p.m., TGU at 15:24 p.m., and TBR at 15:35 p.m., with different rain densities; 2.8 mm, 2.1 mm, and 1.5 mm, respectively. Hence, we take a closer look at their separation distance.

The distance between TGU-TBR and TGU-CH is equivalent, which is 4.14 km. However, only TGU and CH received similar rain rates, with TGU exhibiting a bit less amount of rain at each time than those in CH, while TBR experienced far less rain than both sites. Therefore, the distance between sites appears insignificant in defining the value of diversity gain. Since the observation is during the northeast monsoon, the wind might blow from the northeast of TGU with a certain spatial horizontal distance that did not reach TBR, or at least in this case below 4.14 km. This result is consistent with [13] which describes the prevailing wind motion lengthening the rain cell’s structure. The northeast wind brings the moisture from one location to form another cloud (as was also observed in [29]), hence causing rain at another location along its direction (northeast direction) viewing from the TGU.

Meanwhile, in Sepang, the highest rain rate in November 2014, which was PNS, received the peak rainfall of 2.9 mm on 5 November 2014. Consequently, the rest of the investigated areas (PI, LTM, and LBC) were observed on the equivalent day and concurrent time, which was from 15:00 p.m. to 15:50 p.m., as shown in Figure 9. From the figure, PNS received the rainfall earlier than other sites, reaching its peak at 15:00 p.m. with 2.9 mm. LBC, located at the west of PNS, 12.96 km apart, experienced the highest volume apparently, amounting to 3.0 mm but a bit later at 15:37 p.m.

In the meantime, PI, at 8.28 km far southwest of PNS, received a medium rainfall amounting to 0.5 mm only around 15:20 p.m. after heavy rainfall in PNS. LTM is exactly to the south of PNS and recorded nearly no rain despite being separated only 5.56 km from the latter. The PNS rain-event range at this time frame was from 15:00 p.m. to 15:50 p.m., LBC from 15:30 p.m. to 15:50 p.m., and PI from 15:19 p.m. to 15:50 p.m.; each differed in time. Hence, from the view of the PNS, it is expected that the northeast wind blow did not reach LTM. This situation is consistent with the sites in Gombak, described beforehand, that the location along the direction of the wind, including the one to the exact west of a host experiencing similar rainfall, induced by the cloud formation brought by the wind. It was observed that from the wind direction, the rain-cell extent of heavy rainfall is at least 12.96 km, consistent with the analysis by Yeo et al. in [13]. Another option for an effective diverse site is one that is at least 5 km away (an estimation between 4.14 km and 5.56 km), ranging from 90° to 180° (east to south direction) from the due north of a host.

By referring to Figure 6 and Figure 7, in November 2015, all sites in the Gombak district experienced heavy rainfall; none had too little rain, as was depicted in November of the year 2014. The rain rate at 0.01 percent time exceedance in CH was 116.5 mm/h, TBR was 120 mm/h (both in Figure 6), TDK was 86.5 mm/h, and TGU was 130 mm/h (both in Figure 7). In midyear 2015, the El-Niño phenomena started to affect the amount of average rain in Malaysia, which brought more rain than usual. El-Niño arises when warm water heats the atmosphere in the Eastern Pacific, allowing more moisture in the air which then develops into rainstorms [30,31]. Despite receiving heavy downpours in November 2014, TDK experienced the lowest rain rate at 0.01 percent time exceedance in November 2015. Meanwhile, TGU received the highest rain rates at the equivalent percentage of the time. Hence, the TGU diurnal rain event was extracted, and it was observed that the highest rain amount was 2.8 mm on 26 November 2015. Therefore, the rain event in the time frame between 15:00 to 15:50 was presented in Figure 10, together with other Gombak sites (CH, TDK, and TBR) at equal dates and times.

Figure 10 exhibits that TGU has a convective rainfall from 15:00 p.m. to 15:20 p.m. before it reduced to a smaller amount until 15:50 p.m. The TBR rain-event range was similar but commenced a bit delayed, which was at 15:02, and ended earlier at 15:45 p.m. CH received rain eight minutes later than the TGU, and the rain settled down around the same time, 15:50 p.m. TDK was receiving less rain, which amounted only 0.7 mm during its peaks at 15:00 p.m., and ended earlier at 15:17 p.m.; the time where every site was still receiving plenty of rain. The northeast monsoon merged with the El-Niño wind was expected to aggravate the rain density in this area. From the observation, although each site received high rain rates during this month, the rainfall was occurring at different time intervals. Therefore, site diversity could take place, taking advantage of the non-homogeneity of the rain types. TDK is at the southwest of TGU and TBR, separated by 5.24 km and 7.63 km respectively, and only 1.85 km south of CH. Having a moderate rain rate during the month, TDK was expected to be experiencing a lower signal attenuation than other sites during that specific time; hence, this could represent an appropriate diverse site during this spell.

For the Sepang district, Figure 6 shows that the rain rate at PI in November 2015 for 0.01 percent of time exceedance was 108 mm/h and LTM was 98 mm/h; while in Figure 7, the rain rates at LBC for 0.01 percent of time exceedance were the highest among the other sites, which was 125 mm/h, and PNS was 108.5 mm/h. LBC received the highest rain, which amounted to 4.4 mm, on 26 November 2015. Therefore, the rain density in the time series of each site was extracted on that date at a time frame of 16:15 p.m. to 16:55 p.m., as shown in Figure 11.

The rain event was extracted at a different time than the Gombak sites because the evaluation for the diverse site and rain-cell span was confined to the Sepang sites only. From Figure 11, it was observed that LBC started to receive an abundance of rain from 16:17 p.m. to 16:36 p.m. and ended at 16:53 p.m. PI received moderate rainfall five minutes later, with the rainfall peak difference between PI and LBC being 2.8 mm (4.4 mm in PI and 1.6 mm in LBC) until the rain settled down around 16:55 p.m. PNS was also receiving less of a rain amount, which began at 16:22 p.m., with the rainfall peak of 1.3 mm at 16.47 p.m., and ended around 16:51 p.m. Meanwhile, the rain event at LTM was much lower, with a slight peak of 0.1 mm, only in a short duration; from 16:20 p.m. to 16:25 p.m. (5 min).

Since Figure 11 shows the rain events at each site occurred at a different time interval, a site with a severe attenuation signal caused by heavy downpours could be backed up by another clear weather receiver location, hence bringing out possible applications, e.g., resource sharing between those. Considering the PNS area viewpoint, LBC is at its west, so does PI, and LTM is at its south. In November 2015, PNS was no longer experiencing the highest rain rate reading at 0.01% of time exceedance; instead, LBC did. However, no sites received less rain this time. Those rain rates almost equally fall into the heavy downpours categories, showing the impact of both the northeast monsoon and El-Niño wind, which is a phenomenon that happens once every four to seven years.

The El-Niño brought La-Niña, a drier wind cyclone to Malaysia in 2016. La-Niña brings less of a rain amount throughout the country, even in the world. Gombak and Sepang sites showed lower rain rates at 0.01 percent time exceedance in November 2016 than in November of the years before (refer to Figure 6 and Figure 7). In Gombak in November 2016, TDK received rainfall, with a rain rate at 0.01 percent of time exceedance, which amounted to 83 mm/h, CH was 100 mm/h, TGU was 78 mm/h, and TBR was 74 mm/h. November of this year showed that CH received the highest rain rate and its rain-density data in time series were observed. CH experienced peak rainfall of 2.2 mm on 8 November 2016. Therefore, the TDK, TGU, and TBR sites were also observed during the equivalent date specifically from 15:15 p.m. to 15:50 p.m., as shown in Figure 12.

Figure 12 shows that the rainfall occurred at different time durations at each site. In CH, the rainfall duration was from 15:17 p.m. to 15:41 p.m. with peak rainfall at 15.29 p.m., while at TGU, it was from 15:26 p.m. to 15:47 p.m., which was a delay of around ten minutes than the event commencement at CH and seven minutes after the rainfall at CH settled down. While CH was receiving its peak rain, TGU’s rain density was 0.6 mm, and TBR and TDK did not receive any rain. TGU peak rainfall density was 1.6 mm at 15.40 p.m. while, at the same time, CH was receiving only 0.1 mm of rain, and TBR and TDK were receiving none. TBR’s rain event was of a very short duration, which was three minutes and TDK was not experiencing any recorded rain throughout the day.

For the Sepang sites, Figure 6 exhibits that PI was receiving a rain rate of 76.5 mm/h, LTM was 94.5 mm/h; while in Figure 7, the LBC was 79 mm/h and PNS was 83 mm/h at 0.01 percent of time exceedance during November 2016. LTM received the highest rain rate; thus, the rain event in the time series of this site was extracted. It was identified that LTM received a peak rainfall of 2.4 mm on 11 November 2016. Hence, the other sites’ rain-density data in time series during that date and time frame from 15:00 p.m. to 15:50 p.m. were analyzed accordingly, as displayed in Figure 13.

By observing the figure, the rain event at PI and LBC both commenced at the equivalent duration, but one with no rain and the other with a peak rainfall of 0.7 mm at 15:01 p.m., respectively. LTM experienced heavy rainfall around 28 min later than the LBC, with a peak rainfall of 2.4 mm at 15.33 p.m. The LTM rain-density graph demonstrated the downpours at this place were longer in duration, which was from 15:28 to 15:50 than at PNS which started to receive less rain at a delayed time, 15.39 p.m. onwards in this time frame. The rain rates at each site in November 2016 were a bit less than the year before, 2014 and 2015.

### Diversity Gain

From the abovementioned definition, determining the diversity gain requires the attenuation values of each site at a certain time percentage. Each site was paired with another to form a joint attenuation, thus construing 36 sets of gain values for three years. From the observation, a high gain was obtained when comparing sites with the highest rain rate with the lowest one. It is because the rain rate is the significant factor in determining the predicted attenuation by the ITU-R model. Therefore, to simplify the findings, sites with the highest rain rates in November of each year in Gombak and Sepang were chosen as the ones that needed a backup signal, which is the host. The remaining sites were considered diverse locations, hence the gain was calculated, as displayed in Table 4.

In the table, since TDK received the highest rain rate at 0.01 percent time exceedance in November 2014 than the other sites, it was considered as the host and paired with the other remaining sites within the Gombak district to construct the joint attenuation, then the diversity gain was obtained. TDK with 147 mm/h showed the highest gain; 28.24 dB when it was paired with TBR with a rain rate of 31 mm/h even though the distance between them is 7.63 km. The diversity gain obtained by pairing with CH and TGU was not much different. This is consistent with the rain event in Figure 8, which demonstrated that TBR and TDK experienced different rain densities within the same duration. In Sepang, during November of the year 2014, the rain rate at PNS was the highest among other sites, hence contributing to a high gain when paired with LTM (26 mm/h) of 5.56 km, which was 29.35 dB than LBC with further distance, 12.96 km, and PI at 8.28 km away. The observation of time-series rain density in Figure 9 pronounced that LTM did not record any rain within the rain-event duration of PNS. Therefore, the diversity gain that was observed does not have an impact on the increment of the site-separation distance, instead, it is the local rain rate of the site that influences the gain.

In November 2015, TGU showed the highest gain when pairing with TDK (86.5 mm/h) of 5.24 km than other sites, with 8.96 dB; just take note that TGU to CH is 4.14 km, so is TGU to TBR. By referring to Figure 10, TDK was observed to receive less rain than TGU within the same duration. While in Sepang, LBC delivered a high gain when paired with LTM (98 mm/h) of 14.13 km, which received the lowest rain among other sites, as exhibited in Figure 11, within the same duration as LBC. CH in the year 2016 delivered a high gain when paired with TBR (74 mm/h) of 7.4 km, which was 6.04 dB, and LTM in Sepang exhibited a higher gain when paired with PI (76.5 mm/h) of 7.63 km, which was 4.23 dB.

## 4. Discussion

The rain-event analysis of eight sites in Gombak and Sepang reveals that the rain occurrences at a location are subject to the wind direction. In the event of heavy rain at one site, the rain-cell distance was at least 15 km long in the wind direction and at least 5 km to the east during the normal condition of a northeast monsoon. There is a circumstance where the distances are the same, but the site parallel with the wind direction received less rain than the one within the wind direction. The situation was observed at the Gombak sites during the northeast monsoon spell in November 2014, presented in Figure 14.

TGU experienced a high rain rate during the period, and so did CH and TDK, except for TBR. The northeast-wind-caused rain may have a horizontal extent below the distance of TGU to TBR, which is 4.14 km. Hence, the location of TBR was seemingly untouched by the wind, therefore causing less rain. Meanwhile, the same prevailing wind reached CH and TDK, which are located at a similar distance (for CH) but southwest to that of TGU. The baseline orientation angle between TGU and TBR is 0°, TDK to TBR is 43°, and CH to TBR is 29.5°. This sample suggests that baseline orientation angles do not influence the gain; instead, the location of a good diverse site is at least 5 km to the east of the host, within 90° to 180° from the due north (0°), viewed on the map. This conclusion is consistent with the findings in [13].

In November 2014, Sepang sites revealed that the least rain was experienced at LTM. Considering PNS as a host because of its heavy rain at that time, LTM was located right to its south (at 180°), as shown in Figure 15. Despite the site-separation distance of 5.56 km only, the heavy rain caused by the northeast wind at PNS did not affect LTM. Hence, this circumstance suggested that the heavy downpours rain-cell limits are lower than the site-separation distance between PNS and LTM. PI is located 8.28 km southwest of PNS, in the opposite direction of the northeast wind, and it received a higher rain rate (111 mm/h) than the LTM (26.5 mm/h). The baseline orientation angle of LTM relative to the azimuth line is 60.5° concerning PNS, while PI and LBC are 6.5° and 10.5°, respectively. LBC is right to the west of PNS with 12.96 km away and it was also affected by the northeast wind but with less rain than PI. This sample shows that the wind brings moisture and causes heavy rain to LBC and PI of far distance, e.g., 12.96 km, while shorter distances did not affect. However, LTM is right under PNS; therefore, it is consistent with the previous sample that the same prevailing wind could not reach locations of distance 5 km at 90° to 180° from the due north of a host, which means the range angle of the best diverse location is from the east of host till its south, regardless of the baseline orientation angle.

During the El-Niño and La-Niña phenomena in the years 2015 to 2016, all sites were receiving higher and less rain volume than in 2014, respectively. In this case, the diversity gains are high at the diverse location farther than the host, beyond the rain-cell extent. It is because the wind is stronger than normal, bringing more moisture and forming more clouds; hence, the site nearby is also impacted by the heavy downpours. Therefore, to cater to both normal and abnormal years, the site-separation distance is suggested to be extended to 15 km away, from 90° to 180° from due north of the host. However, more data should be observed; therefore, this preliminary suggestion is valid based on the current duration of the rain event and in specific locations.

### 4.1. Model Comparisons

For model comparisons, examples of diversity gain from the paired sites corresponding to November 2014 rain rates were compared with the Hodge, Panagopoulos, Semire, and Yeo models. The pair of PNS–LTM in Sepang was chosen because, in November 2014, PNS received the highest rain rates, while LTM was the lowest. Similar reason applied for TDK–TBR in Gombak. The details of the input used in the comparison are listed in Table 5. The gain was compared from 1 dB to 24 dB, to show the trend of each model. The outcome of the comparisons is shown in Figure 16 and Figure 17 for the PNS–LTM and TDK–TBR gains, respectively.

In Figure 16, it was observed that the Hodge, Panagopoulos, Semire, and Yeo models exhibited higher values than the diversity gain calculated at PNS–LTM. The Hodge model showed a much steeper slope than other models as the attenuation increased. This is due to the model being developed based on temperate-region data, which experiences stratiform rain that occurs in a longer span; therefore, site-separation distance was formulated to have less impact on the predicted gain. Semire and Panagopoulos models’ graphs were in parallel with each other, with the predicted gain in a higher value than the one obtained at PNS–LTM. The Semire model was developed based on a tropical regions’ dataset, with a prediction limitation to below 50° of the elevation angle; while in this investigation, a higher elevation angle was used, which was 68.8°. The Panagopoulos model was developed using temperate-region databases, which normally have longer rain cells. However, the model emphasized the site-separation distance and attenuation influence. Both still give impact to the model as they increase, unlike the Hodge model. Therefore, with-logarithm function in its formula, the lower site-separation distance gives a rapidly higher gain than the longer one. Nonetheless, in this investigation, the condition is not necessarily true because of the wind impact. The Yeo model delivered a similar graph shape with Semire but steeper; however, it overestimated the measured gain at PNS–LTM. The Yeo model was designed with a maximum 50° elevation angle; therefore, it may have a limitation. The higher the elevation angle, the signal is less attenuated because of the short effective length and propagation path. The signal may experience a reduction in a series of rain events compared to the longer path (low elevation angle). Furthermore, the path into the convective rain cell may be shortened.

Figure 17 showed that the predicted gain of the Hodge, Panagopoulos, Semire, and Yeo models overestimated the gain at the TDK–TBR. Panagopoulos’s prediction graph was in parallel with Semire, but with lower values than the TDK–TBR. The prediction of Yeo and Hodge’s model was almost the same higher values, but steeper in slope than the diversity gain at TDK–TBR as the attenuation increased. Hence, none of the models fit the constructed gain of PNS–LTM and TDK–TBR. More observations could be conducted; however, these comparisons are sufficient for the current dataset. However, interested readers may follow the details of the model’s comparison with real time measurement satellite signal in [32] and models’ characteristics in [33].

### 4.2. Limitations of Data

The rain density readings used for the research collected from DID are at least 99.9% available. The readings reported may have some mechanical issues such that the tipping bucket failed to tip during the extremely heavy rain; thus, this might cause discrepancies during data logging. Therefore, during the analysis to determine the rainfall rate at the percentage of time probability, the graph shows an indefinite value as the percentage becomes less. However, the most important thing is that the graph trend can be seen and identified (refer to Figure 6 and Figure 7).

## 5. Conclusions

This article presented a detailed analysis of the rain rate and rain attenuation at Sepang and Gombak sites, in West Peninsular Malaysia using the rain density readings of the year 2014 to mid-July 2017. The impacts of prevailing wind and El-Niño and La-Niña winds were also investigated during November, one of the months in the northeast monsoon. The analysis of the sample proposed that the baseline orientation angle does not affect the diversity gain because of the wind impact. The sites with a distance of at least 15 km to the east of the host, and in the range from 90° to 180° from its due north exhibited the higher gain. It is because Malaysia, and perhaps, countries located nearby, experience the northeast monsoon, with the northeast wind blowing clouds and consequent rain into the location of its direction, causing similar rain-induced attenuation, thus bringing lower gain to the respective host. Cross-district analysis at a site-separation distance of more than 40 km does not suggest a higher gain. Local rain may exhibit attenuation dissimilarity more than the site at far. Therefore, lengthening the site-separation distance has less impact on the gain, and the cost of the infrastructure could be decreased (in terms of underground cable if needed). The findings of this analysis could be used to reformulate the site diversity and gain empirical formulas specific to the region that is affected by the season. This work of multisite diversity performance is also beneficial in resource-sharing applications in the future.

## Figures and Tables

**Figure 1 sensors-23-06424-f001:**
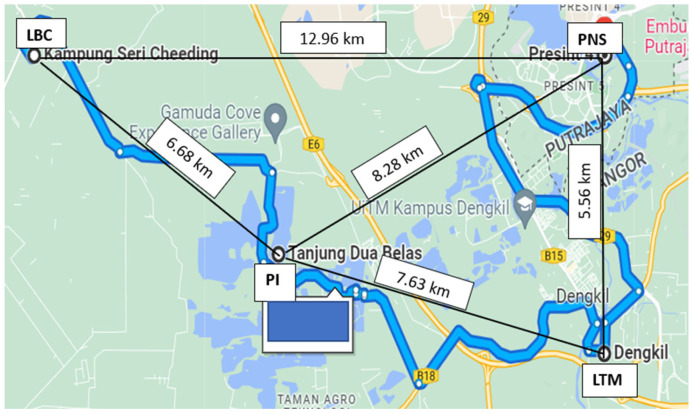
The locations of four investigated sites in Sepang districts.

**Figure 2 sensors-23-06424-f002:**
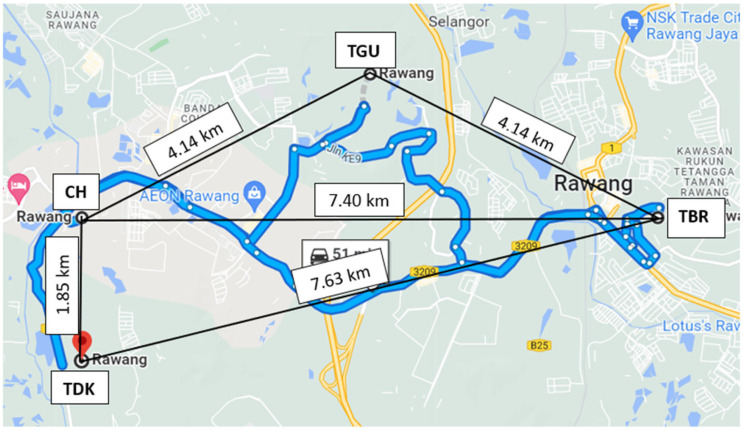
The locations of four investigated sites in Gombak districts.

**Figure 3 sensors-23-06424-f003:**
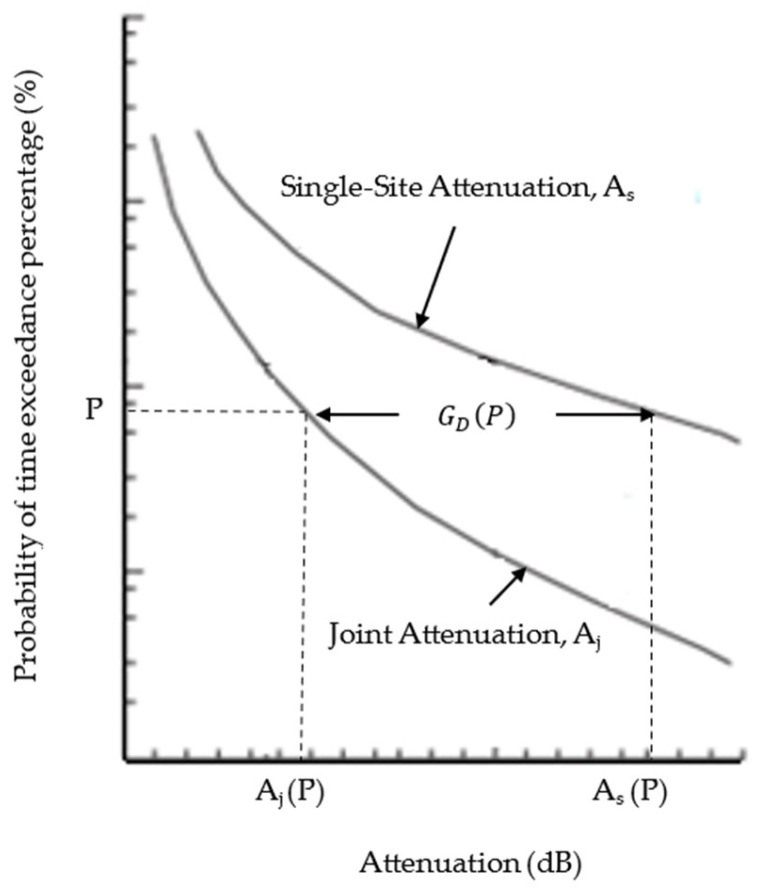
Definition of diversity gain [28].

**Figure 4 sensors-23-06424-f004:**
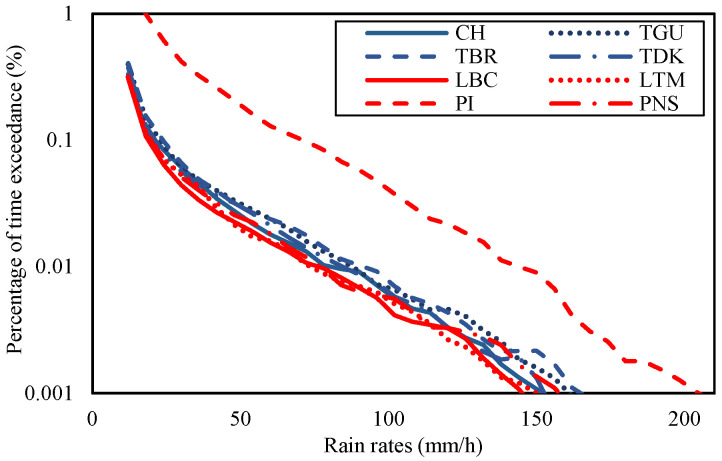
Rain rates at CH, TGU, TBR, and TDK in Gombak and LBC, LTM, PI, and PNS in Sepang from 2014 to 2017.

**Figure 5 sensors-23-06424-f005:**
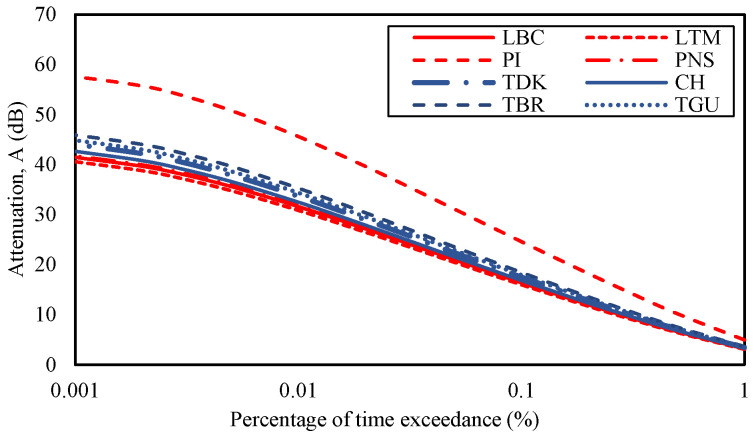
ITU-R rain attenuation modeling of each site.

**Figure 6 sensors-23-06424-f006:**
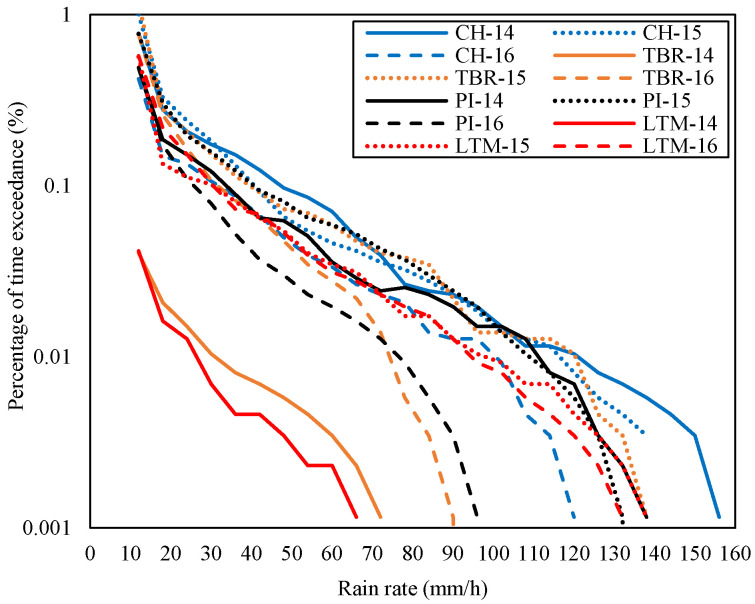
Rain rates at TBR and CH of Gombak, PI, and LTM of Sepang in November of each year (2014–2016).

**Figure 7 sensors-23-06424-f007:**
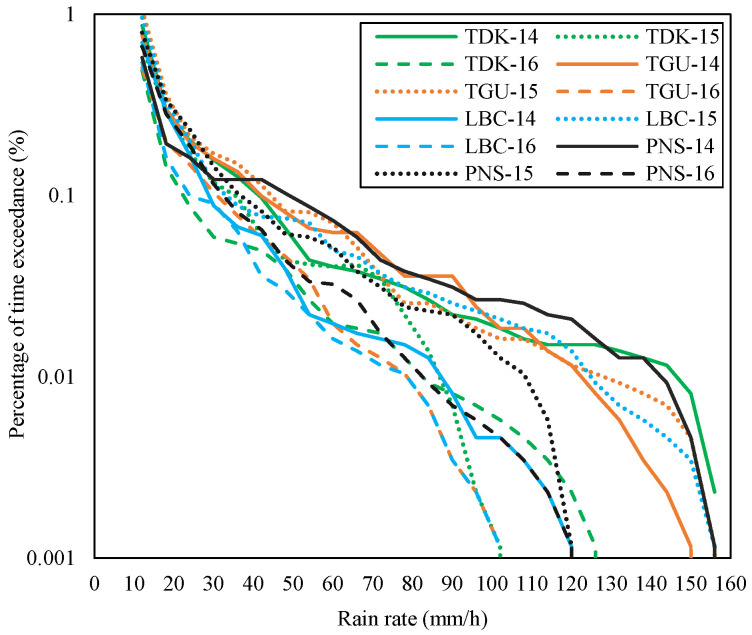
Rain rates at TDK and TGU of Gombak, LBC, and PNS of Sepang during November of 2014–2016.

**Figure 8 sensors-23-06424-f008:**
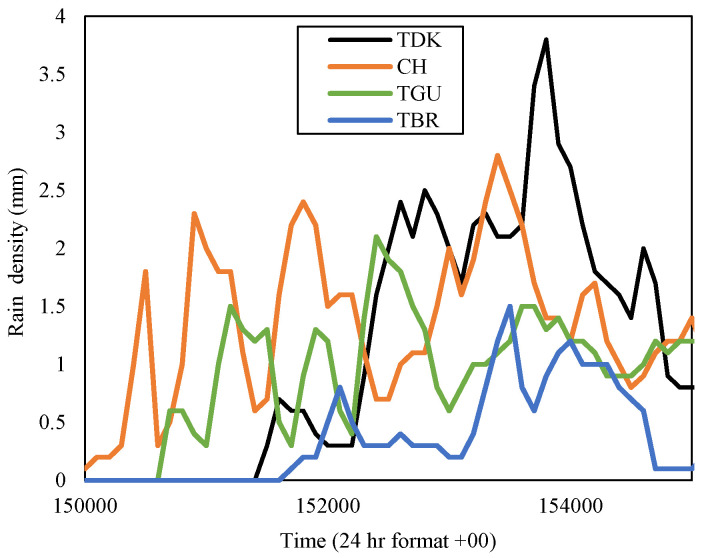
Rain event from 15:00 to 15:50 on 21 November 2014, at Gombak sites.

**Figure 9 sensors-23-06424-f009:**
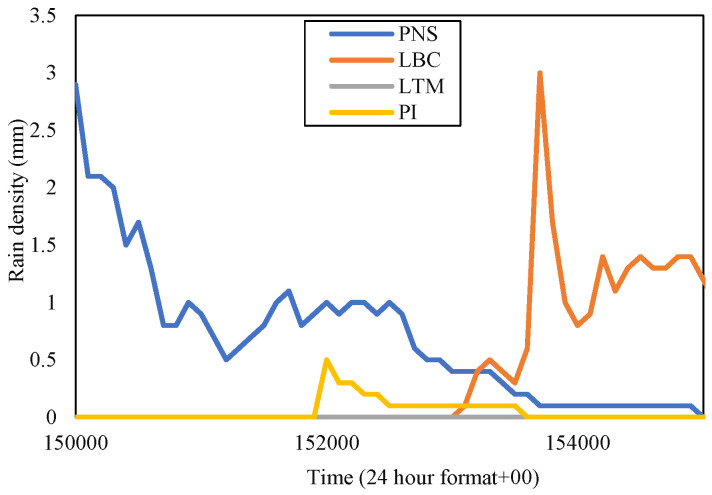
Rain event from 15:00 to 15:50 on 5 November 2014 at Sepang sites.

**Figure 10 sensors-23-06424-f010:**
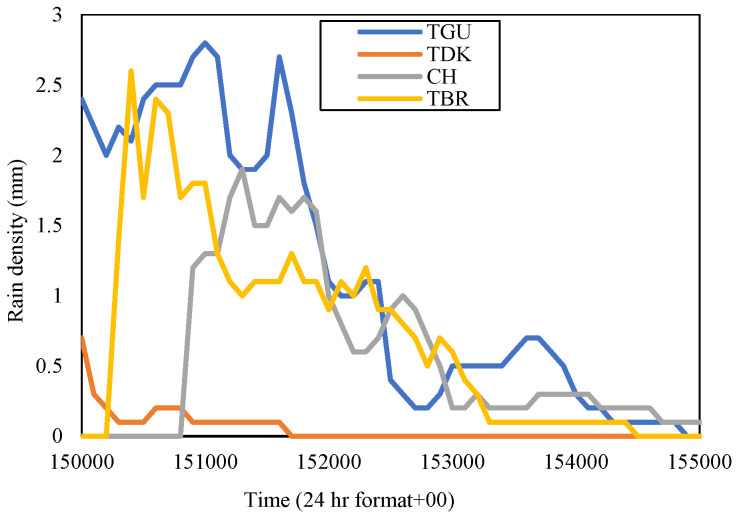
Rain event from 15:00 p.m. to 15:50 p.m. on 26 November 2015, at Gombak sites.

**Figure 11 sensors-23-06424-f011:**
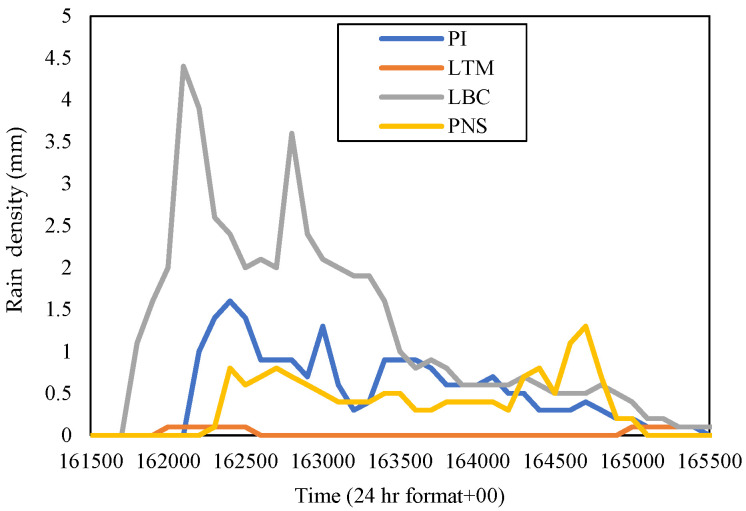
Rain event from 16:15 to 16:55 on 26 November 2015, at Sepang sites.

**Figure 12 sensors-23-06424-f012:**
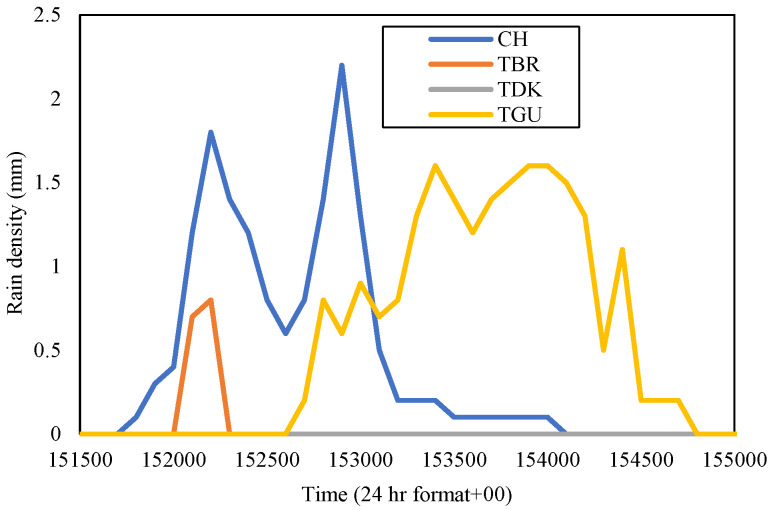
Rain event from 15:15 to 15:50 on 8 November 2016, at Gombak sites.

**Figure 13 sensors-23-06424-f013:**
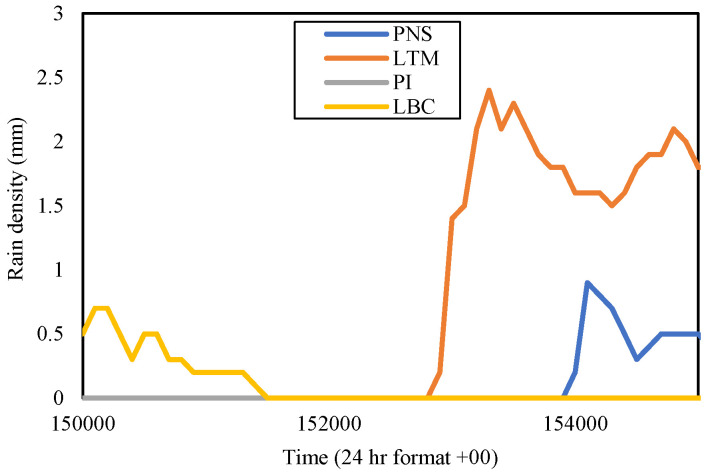
Rain event from 15:00 to 15:50 on 11 November 2016, at Sepang sites.

**Figure 14 sensors-23-06424-f014:**
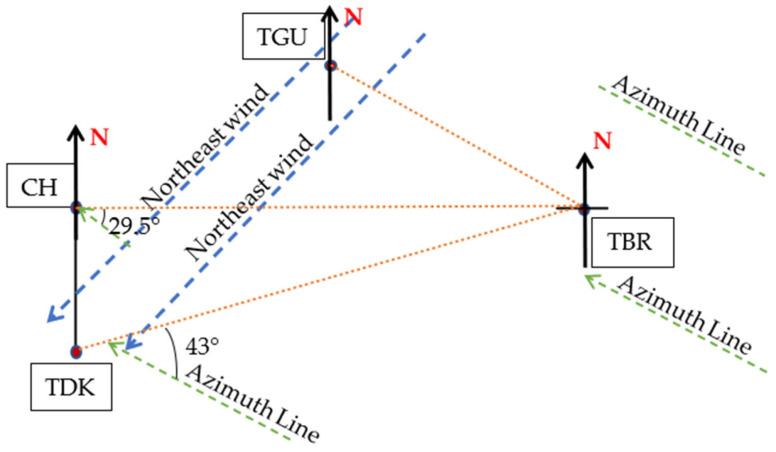
Northeast wind rain affects Gombak sites.

**Figure 15 sensors-23-06424-f015:**
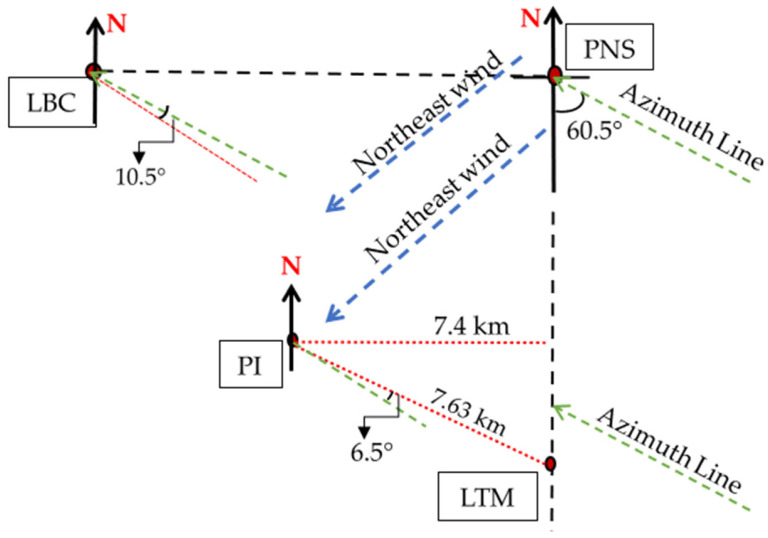
Northeast wind rain effects on Sepang sites.

**Figure 16 sensors-23-06424-f016:**
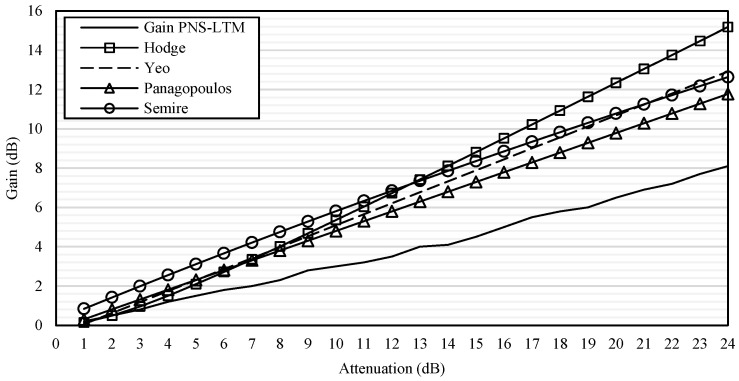
Gain of PNS–LTM compared with Hodge, Yeo, Panagopoulos, and Semire models.

**Figure 17 sensors-23-06424-f017:**
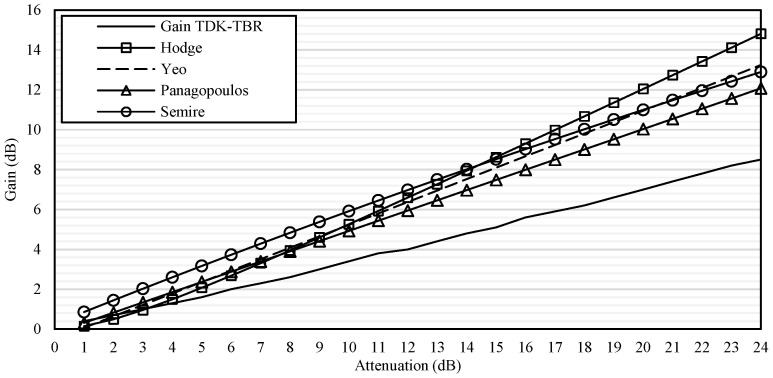
The gain of TDK–TBR compared with Hodge, Yeo, Panagopoulos, and Semire models.

**Table 1 sensors-23-06424-t001:** Site-separation distances each district.

Sepang	Distance (km)	Gombak	Distance (km)
LBC–PNS	12.96	CH–TGU	4.14
LBC–LTM	14.13	CH–TBR	7.40
LBC–PI	6.68	CH–TDK	1.85
LTM–PNS	5.56	TDK–TGU	5.24
PI–PNS	8.28	TDK–TBR	7.63

**Table 2 sensors-23-06424-t002:** Gombak–Sepang Distance.

Gombak–Sepang	Distances (km)	Gombak–Sepang	Distances (km)
CH–PI	51.26	TBR–PI	50.17
CH–LTM	52.84	TBR–LTM	53.07
CH–LBC	46.46	TBR–LBC	46.37
CH–PNS	49.89	TBR–PNS	47.75
TGU–PI	52.41	TDK–PI	49.45
TGU–LTM	55.75	TDK–LTM	53.35
CH–PI	51.26	TBR–PI	50.17

**Table 3 sensors-23-06424-t003:** Diversity gains of PI as the host.

Sites (PI–Other Sites)	Gain (dB)	Distance (km)
PI–LBC	14.01	6.68
PI–LTM	14.77	7.63
PI–PNS	13.76	8.28
PI–TDK	11.96	49.45
PI–CH	13.14	51.26
PI–TBR	10.35	50.17
PI–TGU	11.26	52.41

**Table 4 sensors-23-06424-t004:** Site diversity gain in Gombak and Sepang from 2014 to 2016.

Year	District	Site (Rain Rate)	Host–Diverse	Gain (dB)
2014	Gombak	TDK (147 mm/h)	TDK–CH	4.91
			TDK–TBR	28.24
			TDK–TGU	4.53
	Sepang	PNS (142.5 mm/h)	PNS–PI	5.93
			PNS–LTM	29.35
			PNS–LBC	11.13
2015	Gombak	TGU (130 mm/h)	TGU–CH	2.57
			TGU–TBR	1.89
			TGU–TDK	8.96
	Sepang	LBC (125 mm/h)	LBC–PI	3.35
			LBC–LTM	5.46
			LBC–PNS	3.25
2016	Gombak	CH (100 mm/h)	CH–TBR	6.04
			CH–TDK	3.83
			CH–TGU	5.04
	Sepang	LTM (94.5 mm/h)	LTM–PI	4.23
			LTM–LBC	3.61
			LTM–PNS	2.65

**Table 5 sensors-23-06424-t005:** Input of each model.

Input	PNS–LTM (PNS Host)	TDK–TBR (TDK Host)
Frequency, *f* (GHz)	20.2	20.2
Elevation Angle, *θ* (°)	68.8	68.8
Distance, *d* (km)	5.6	5.85
Baseline Angle (°)	60.5	43

## Data Availability

The rain density and satellite-signal measurement raw data were taken from DID, Selangor, Malaysia, and MEASAT Satellite Systems Sdn. Bhd. respectively, and it is confidential.

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
