# Peer review of "The Horizontal Rain-Cell Span and Wind Impact on Multisite Diversity Scheme in a Tropical Region during El-Niño and La-Niña"

_sensors, 2023, doi:10.3390/s23146424_

Round 1

Reviewer 1 Report

This article presented the rain analysis from the year 2014 to mid-July 2017 at eight sites in the Gombak and Sepang districts of Malaysia. The rain rates of 0.01 percent of time exceedance were extracted from the analysis, which was then used to predict the attenuation using the ITU-R P.618-13 rain attenuation model. The results of this study may contribute to evaluate the inter and cross-district gain characteristics. However, there are some concerns that the authors should address before it can be considered for publication.

(1) The abstract should clearly state why this study was conducted and the significance of the research.

(2) The last paragraph of the introduction is too detailed, and the authors can simply explain the research data, methods, content, and significance.

(3) Are there missing data in the rain density dataset? Usually the missing rain data may affect the results of this study (ie., Shen et al., 2014), which should be discussed in the limitation of this study.

(4) In order to further highlight the innovation of this article, it is better to compare the results of this study with some other studies.

(5) There needs to be a comprehensive discussion and interpretation of the results. The current discussion chapter is inadequate.

(6) A paragraph of limitation discussion should be added to clarify the limitation or uncertainty of data and methods in this current study.

Reference:

Spatiotemporal change of diurnal temperature range and its relationship with sunshine duration and precipitation in China, 2014, 119, 13163-13179.

Reviewer 2 Report

Comments:

  1. The title of the paper is informative and clearly states the focus of the study. It accurately reflects the content of the article.
  2. The authors provide a comprehensive abstract that summarizes the main objectives, methodology, and findings of the study. The abstract provides a clear overview of the research, making it easy for readers to understand the context and importance of the study.
  3. The introduction of the paper provides a good background and rationale for the study. It explains the significance of diversity gain in a site diversity scheme and highlights the factors that influence it. However, the introduction could benefit from a clearer statement of the research objectives or research questions that the study aims to address.
  4. The methodology section is well-detailed and provides the necessary information for readers to understand how the study was conducted. The use of the ITU-R P.618-13 rain attenuation model is appropriate for predicting attenuation based on rain rates. However, it would be helpful if the authors provided more information on the data collection process and the specific variables measured or recorded at each site.
  5. The results and discussion section provides a thorough analysis of the rain data and its impact on diversity gain. The findings regarding the pairing of high rain rate sites with lesser ones and the influence of wind direction on rain-cell extent are interesting. However, more specific quantitative results and statistical analysis would enhance the robustness of the study.
  6. While the paper presents valuable insights into the horizontal extent and wind impact on multi-site diversity schemes in tropical regions, there are a few areas that need improvement:

a. The paper lacks a clear research hypothesis or research questions. It would be beneficial to explicitly state the research objectives at the beginning of the paper.

b. The paper would benefit from more detailed information on the experimental setup and data collection procedures. This would help readers understand the reliability and generalizability of the findings.

c. The discussion should include a comparison of the study's results with existing literature or theoretical models to support the findings and provide a broader context.

d. The conclusions should provide a concise summary of the study's key findings and their implications for future research or practical applications.

  1. Overall, while the paper addresses an important topic and presents interesting findings, it falls short in several aspects. Given the limitations mentioned above, I recommend rejecting the paper in its current form. However, I encourage the authors to address the mentioned points and resubmit their work for further consideration.

Minor editing of English language required

Round 2

Reviewer 2 Report

Thank you for revising the manuscript as per the suggested comments. I appreciate your efforts in addressing the reviewer's comments and making the necessary revisions to improve the quality and clarity of the manuscript. Your dedication to enhancing the content is commendable. I have reviewed the revised version and I am pleased to inform you that the article may be accepted in its present form.